# Flood relief logistics planning for coastal cities: a case study in Shanghai, China

Pujun Liang[1], Jie Yin[1, 2, 3]*, Dandan Wang[4], Yi Lu[5], Yuhan Yang[1], Dan Gao[1], Jianfeng Mai[1]

[1]School of Geographic Sciences, East China Normal University, Shanghai 200241, China

[2]Key Laboratory of Geographic Information Science (Ministry of Education), East China Normal University, Shanghai 200241, China

[3]Research Center for China Administrative Division, East China Normal University, Shanghai 202162, China

[4]National Disaster Reduction Center of China, Ministry of Emergency Management of People's Republic of China, Beijing 100124, China

[5]Taizhou Key Laboratory of Typhoon and Marine Meteorology, Taizhou 318000, China

* Correspondence : Jie Yin (jyin@geo.ecnu.edu.cn)

**Abstract:** Coastal cities are becoming more vulnerable to flood risks due to climate change, rising sea levels, intense storm surges, population growth, and land subsidence. Developing emergency preparedness and response strategies can reduce the impact of coastal flooding and improve a city's resilience. This article presents a flood relief logistics planning approach aimed at providing decision-makers with a feasible framework. The framework integrates geographic information system (GIS) network analysis and resource allocation optimization models. Considering the equity of resource allocation, a bi-objective allocation model that minimizes the total transportation cost and maximum unsatisfied rate is developed. This flood relief logistics planning approach is applied to Shanghai, China, to present feasible distribution strategies. The case study indicates that the current spatial distribution of Emergency Reserve Warehouses (ERWs) and Emergency Flood Shelters (EFSs) in Shanghai may be vulnerable to extreme flood events. Under a 1000-year coastal flood scenario, the existing emergency resources are insufficient to meet the needs of the affected elderly population. In situations of resource scarcity, reducing the maximum unsatisfied rate can help improve the equity of resource allocation. Furthermore, incorporating private warehouse clubs (WHCs) into government emergency logistics through public-private collaboration could reduce governmental burden and improves system efficiency and resilience. This study provides a scientific reference for developing flood relief logistics plans in Shanghai, and it presents a transferable framework that is applicable to other coastal cities.

**Keywords**: Coastal flooding; Flood relief logistics planning; Relief distribution; NSGA-II; Bi-objective allocation model; Emergency flood shelters; Emergency reserve warehouses.

## 1. Introduction

Flooding is among the most frequent and catastrophic natural hazards, causing substantial global casualties and losses. Over the past two decades, the number of major flood events has more than doubled, claiming approximately 1.2 million lives and impacting over 4.03 billion people (Mizutori and Guha-Sapir, 2020). In particular, coastal cities, where population and assets are concentrated, have been severely affected by storm-induced flooding(Cook and Merwade, 2009). For example, Hurricane Katrina-induced flooding overwhelmed 80% of New Orleans in 2005, resulting in approximately 1833 fatalities, displacing approximately 770,000 residents,

and causing over $100 billion in losses (Kates et al., 2006; Townsend, 2006). Hurricane Sandy, which made
landfall near Atlantic City in 2012, triggered a catastrophic storm surge in New York City, causing disruptions of
city systems, 43 deaths, and approximately $19 billion in losses (Bloomberg, 2013). Typhoon Mangkhut, which
hit Hong Kong in 2018, generated a record-breaking storm surge that caused widespread damage across the city.
At least 458 people were injured, and the direct economic losses amounted to approximately HKD 4.6 billion
(Choy and Wu, 2018). Although the threat is already considerable, various ongoing trends could greatly amplify
flood risks in the coming years. Sea level rise (SLR) and storm surge intensification, driven by climate change,
exacerbate the risk of coastal flooding (Field et al., 2014), and the frequency of coastal flooding is predicted to
double in the next few decades (Vitousek et al., 2017). Moreover, the trend of rapid urbanization in coastal zones
is expected to continue in the future (Hallegatte et al., 2013), which may further amplify the impact of flooding
(Jongman, 2018). Disaster risk management systems face increasing challenges in adapting to evolving risk
profiles (IPCC, 2023). Hence, as the most crucial pathway for enabling governmental actors to respond to this
threat, the operational implementation of coastal flooding adaptations in emergency management must be
examined.
In recent decades, the increase in the frequency and intensity of coastal flooding has attracted growing attention
from the public, researchers and decision-makers worldwide (Moftakhari et al., 2017; Couasnon et al., 2020). In
response, society has sought to address this threat, with global policy frameworks and national strategies gradually
emerging to reduce coastal flooding risk. In the 21 century, disaster preparedness and emergency response have
been emphasized as the priorities of the Hyogo Framework for Action 2005-2015 and the Sendai Framework for
Disaster Reduction 2015-2030, respectively (ISDR, 2005; UN/ISDR, 2015). Moreover, the New Urban Agenda
outlines actions to strengthen cities' capacities to reduce disaster risk and mitigate their impacts (Habitat III, 2017).
The Making Cities Resilient 2030 (MCR2030) initiative advocates for incorporating climate risk projections into
disaster risk reduction and resilience strategies (UNDRR, 2022). Yin et al. (2024) demonstrated the improved
performance of risk-informed, strategic evacuation planning in advance of coastal flooding.
In disaster management, disaster relief logistics are essential to save human lives and reduce damage (Qin and
Liu, 2017). However, disaster relief logistics systems have many deficiencies, such as resource shortage,
ineffective communication regarding supply and demand information, and improper allocation of resources(Hu
et al., 2019; Zhang, 2016). During Hurricane Katrina, numerous American citizens experienced profound levels
of distress while ensconced in shelters, devoid of sufficient provisions (Brodie et al., 2006). In the aftermath of
the Great East Japan Earthquake, the failure of certain systems, unreasonable warehouse placements, and other
issues resulted in the irrational distribution of emergency supplies and a state of chaos (Ranghieri and Ishiwatari,
2014). Thus, developing disaster relief logistics strategies to ensure the availability of adequate supplies and
capacity is essential to prepare for coastal flooding and effectively manage emergencies.
The mechanism of disaster relief logistics plays a vital role in ensuring the efficiency of emergency response
efforts (İvgin, 2013). In past decades, numerous studies have been conducted in the field of disaster relief logistics,
with the majority focusing on developing mathematical optimization models to solve this problem (Rawls and
Turnquist 2010; Wang et al. 2019). Previous relevant research has focused primarily on facility location, stock
prepositioning, and relief distribution (Caunhye et al., 2012; Kundu et al., 2022). For example, Rawls et al. (2010)
developed an emergency response planning tool that uses a two-stage stochastic mixed-integer programming
model to determine the locations and quantities of multiple types of emergency supplies to be prepositioned.
Zhang et al. (2022) proposed a distributional robust optimization model to determine the optimal location of
emergency facilities and resource allocation. Jana et al. (2022) proposed a probabilistic fuzzy goal programming
model for making decisions to manage the supply of emergency relief materials. The goal of disaster relief
logistics decision-making is primarily to improve the effectiveness and fairness of an emergency response. For
instance, Huang et al. (2015) developed a tri-objective allocation network model with a focus on life-saving utility,
delay costs and equity. Additionally, Halit Üster et al. (2017) designed a strategic emergency preparedness
network that aimed to minimize the maximum travel distance for an evacuee and the overall system cost. And,
Ghasemi et al. (2022) proposed a scenario-based stochastic multi-objective model to minimize the expected total
cost, maximum unsatisfied demand for relief personnel, and total probability of unsuccessful evacuations.

Despite considerable research in the field of disaster relief logistics, only a few studies have examined the impact
of floods on resource distribution logistics, particularly the disruptions caused by the inundation of emergency
facilities and roads. For example, Rodríguez-Espíndola et al. (2015) used GIS to create flood maps and developed
a multiobjective optimization model to determine the locations of emergency facilities, assess the allocation of
prepositioned goods, and establish a distribution plan based on flood patterns. Christopher Mejia-Argueta et al.
(2018) evaluated flood hazards using GIS and proposed a multicriteria optimization model that considered
evacuation and distribution flow-time and budget usage to evacuate people and distribute relief supplies.
Additionally, considering uncertainties in the scale and impact of flooding, Garrido et al. (2015) proposed a
stochastic programming model to optimize the inventory of emergency supplies as well as vehicle availability in
flood emergency logistics. However, it is noteworthy that these existing studies were primarily based on historical
flood scenarios and did not adequately consider the potential increased risks posed by extreme flood events under
future climate scenarios. Moreover, most of these studies focused on optimizing the efficiency of resource
distribution, whereas relatively little research has focused on ensuring equity in the allocation process.

In this paper, we introduce a scenario-based approach to flood relief logistics planning for coastal cities through
a combination of GIS analysis and resource allocation models. GIS-based analysis is used to estimate facility
availability and relief resource demand under future coastal flood scenarios. Then, resource allocation models
based on the estimation of supply and demand are developed and implemented to determine the locations of active
warehouses and to support resource allocation planning. This approach has been applied to the metropolitan region
of Shanghai, which is one of the coastal cities that is most exposed to flooding in the world (Balica et al., 2012)
and characterized by a large population and significant aging issues (Liu et al., 2021). This study aims to provide
scientific and technical support for enhancing contingency plans for coastal megacities, and to offer new insights
into operational emergency management decisions for major coastal cities facing significant flood risks. The rest
of this paper is organized as follows. In Section 2, the modelling methodology is presented. Section 3 describes
the case study results. The conclusions are summarized in Section 4.

**2. Methodology**

## 2.1. Problem description

As storm-induced flooding poses a significant threat to coastal cities, emergency authorities may be required to respond by evacuating vulnerable populations to emergency flood shelters (EFSs) and providing them with necessary supplies. In distributing these resources, emergency responders must assess the needs of evacuees to develop an effective allocation plan. In this study, we propose a flood relief logistics planning framework (Fig. 1) that can assist decision-makers in addressing the challenge of distributing resources to vulnerable populations affected by coastal flooding at the city level. The methodology integrates GIS with resource allocation models to present a planning framework based on coastal flood mapping. We evaluate available candidate facilities, the remaining road network and affected people and obtain information regarding basic population needs and the shortest path through GIS analysis. Then, the obtained information is used in allocation models to determine which ERWs to activate and how to supply resources to the activated shelters.

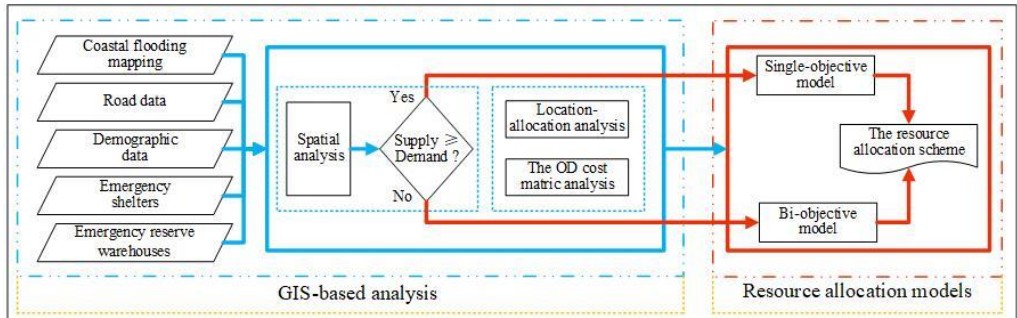

**Figure 1**. The flood relief logistics planning framework.

## 2.2 Assumptions

In the development of the proposed methodology, the following assumptions were made to simplify the model:

A1: The numbers, locations and capacities of emergency facilities such as warehouses and shelters are known.

A2: Pedestrians and vehicles are not allowed to use roads with flood inundation depths greater than 30 cm, which is the common standard for urban roads that are closed due to waterlogging (Yin et al., 2016b).

A3: Flooded EFSs will be set as invalid shelters. ERWs located in areas with flood inundation depths above 30 cm or that cannot be reached by vehicles are considered invalid warehouses.

A4: Only the needs of the elderly population in the affected areas are considered because these individuals are often most vulnerable to flood disasters given their limited ability to acquire information, make rapid judgements and take action.

A5: Resources refer to daily living supplies (e.g., water and food). A kit of goods can meet the requirements of a person during a flood evacuation.

A6: Resources are transported only from ERWs to EFSs and not separately to other areas.

A7: There is no limit on number of vehicles, and each vehicle can transport one hundred kits.

A8: Any resource allocation within the city can be completed within 12 hours. Notably, the 14th Five-Year National Comprehensive Disaster Prevention and Mitigation Plan in China stipulates that the basic living needs of affected individuals will be met within 12 hours.

A9: The affected elderly population is allowed to shelter in advance following a flood warning. The authorities can then establish strategies for distributing goods after evaluating the effects of flooding.

154

**2.3 GIS-based analysis**

Based on coastal flood mapping, the locations of available ERWs and EFSs as well as unsubmerged road networks and affected communities can be determined through GIS-based spatial and network analysis. Additionally, owing to the lack of more detailed spatial distribution data for the elderly population, this study assumes a uniform distribution within the community. The number of elderly individuals in each affected community can be calculated using Equation 1.

$$AP = \frac{IA}{A} * P \tag{1}$$

where AP represents the number of affected elderly people in the community, IA is the inundation area in the community, A is the total area in the community, and P is the total number of elderly people in the community.

164

Then, according to GIS-based location-allocation analysis, the locations of activated EFSs and the number of elderly individuals in the activated EFSs can be identified. Specifically, all available shelters are candidate sites for activated shelters, and all affected community centroids are considered demand points and used to assign affected elderly people in the community to shelters by minimizing the total distance travelled while considering shelter capacity constraints. In this way, we determine the number of affected elderly people at each activated shelter. This information is used to determine how many kits of goods should be distributed to each shelter in subsequent resource allocation models. In addition, the matrix of the shortest path between available warehouses and activated shelters affected by flooding is used as an input for the resource allocation models, and this information is calculated on the basis of origin–destination (OD) cost matrix analysis in GIS with the objective of minimizing the total route length.

175

**2.4. Resource allocation models**

Based on the above description and assumptions, models for resource allocation are designed. The following resource allocation models are used to establish allocation plans and their components, such as the number of activated ERWs that need to provide supplies and the quantity of kits that must be transported from ERWs to activated EFSs. Considering that emergency responders encounter two cases, namely, sufficient or insufficient supplies, a resource allocation model with two case outcomes is established.

182

**2.4.1 Notations and definitions**

The full mathematical model uses the following notation.

**Indices and sets**

$I$: set of available ERWs, indexed by $i \in I$

$J$: set of activated EFSs, indexed by $j \in J$

**Parameters**

$c_{ij}$: the unit transportation cost per unit distance per hundred kits transported between available ERW i and activated EFS j

$d_{ij}$: the shortest path between available ERW i and activated EFS j

$P_i$: the inventory of available ERW i

$Q_j$: the demand of activated EFS j
ω: the lowest satisfaction rate for each activated EFS
**decision variables**
$X_{ij}$: a binary value of 0 or 1, representing whether available ERW i serves activated EFS j
$Y_{ij}$: a nonnegative variable, representing the quantity of allocated resources from available ERW i to
activated EFS j
$R_j$: the satisfaction rate, representing the quantity of elderly individuals receiving supplies as a percentage
of the total elderly population at each activated EFS j

**2.4.2 Sufficient supply scenario**
In a sufficient supply situation, the total available supplies in the city can meet the total demand of the refugee
population during a flood. That is, everyone can obtain sufficient resources. Therefore, we establish a single-
objective allocation model that considers only the efficiency objective. This method aims to optimize system
efficiency by minimizing the total transportation cost. The objective function can be defined as:
$$minf = \sum_{j \in J} \sum_{i \in I} c_{ij} d_{ij} Y_{ij} \tag{1}$$

subject to:
$$\sum_{i \in I} Y_{ij} = Q_j \qquad \forall j \in J \tag{2}$$

$$\sum_{j \in J} Y_{ij} \leq P_i \qquad \forall i \in I \tag{3}$$

Objective function (1) minimizes the total transportation cost. To satisfy the demand of each EFS, the related
constraint function is expressed in Equation (2) to ensure that the resources received are equal to the demand for
each EFS. Equation (3) ensures that the supplies allocated from each ERW are less than its overall inventory.

**2.4.3 Insufficient supply scenario**
During times of catastrophic coastal flooding, we assume that the relief supplies of available ERWs are inadequate
and thus cannot meet the demand of all evacuees. When supply is insufficient to meet demand, emergency
managers should ensure that resources are distributed fairly across regions to avoid the humanitarian inequalities
that result from unbalanced allocation. To consider both efficiency and equity, a bi-objective programming model
with a trade-off between efficiency and equity is established to provide decision-makers with different options for
resource allocation. Specifically, our model includes two possible objectives: objective (f1), which minimizes the
total transportation cost as the efficiency goal, and objective (f2), which minimizes the maximum unsatisfied rate
as the equity goal. The maximum unsatisfied rate refers to the highest unmet demand rate among these regions.
By minimizing this rate, we aim to reduce disparities in unmet demand across regions, ensuring that no area
experiences extreme shortages. The objective function can be defined as:
$$minf1 = \sum_{j \in J} \sum_{i \in I} c_{ij} d_{ij} Y_{ij} \tag{1}$$

$$minf2 = max(1 - R_j) \tag{2}$$

$$\left. \sum_{i\in I} Y_{ij} \middle/ Q_j \right. = R_j \qquad \forall j \in J \tag{3}$$

subject to:

$$\sum_{j\in J} Y_{ij} = P_i \qquad \forall i \in I \tag{4}$$

$$\sum_{i\in I} Y_{ij} \leq Q_j \qquad \forall j \in J \tag{5}$$

$$R_j \geq \omega \qquad \forall j \in J \tag{6}$$

Equation (3) is the formula for calculating the satisfaction rate $R_j$, which equals the total allocated resources as a percentage of the total demand at each EFS. Equation (4) ensures that the resources allocated from ERWs do not exceed the total inventory of ERWs. Additionally, the supplies received should be less than the demand for each EFS, and the constraint function is expressed in Function (5). Function (6) ensures that the minimum satisfaction rate is met for each EFS.

## 3. Case study

### 3.1. Study area

A case study is conducted in Shanghai, China. Shanghai is located on the west coast of the Pacific Ocean and is in the floodplain of the Yangtze River Delta. This city is one of the financial centres of China, and its gross domestic product (GDP) is among the top 10 in the world. Its a total population is over 24 million, and 16.3% of its residents are 65 years old or older (Shanghai Municipal Statistics Bureau, 2021).

Coastal flooding has historically been a frequent issue in Shanghai. For example, Typhoon Winnie in 1997 resulted in RMB 635 million in economic losses and approximately 15000 affected people (Quan, 2014). In response, Shanghai has built emergency shelters and prepared relief supplies in ERWs. By the end of 2020, 117 emergency shelters had been built in Shanghai to provide basic security protection for affected people. This work continues, and the goal of 1.5 m$^2$ of shelter space per capita is expected to be met by 2025 (General Office of the Shanghai Municipal People's Government, 2021). Currently, Shanghai has emergency warehouses at three levels: the city level, district level and township level. There is one city-level warehouse, with the main depot in Jiading District and a branch depot in Minhang District.

### 3.2. Data sources

Data such as flood inundation maps, road data, demographic data, emergency warehouse information, emergency shelter information and warehouse clubs are obtained from different sources for this study (Table 1).

**Table 1.** Data Source Information.

| Data type | Source | Description |
|---|---|---|
| Flood Inundation Maps | Yin et al. (2020) | Simulated coastal flood inundation scenarios for 100-year and 1000-year return periods under the RCP 8.5 scenario. |
| Road Network | Key Laboratory of the Ministry of Education at East China Normal University. | Comprises approximately 243,000 road sections with attributes. |
| Demographic | Shanghai Municipal Bureau of Statistics | Detailed demographic information at the community level. |

| Emergency Warehouse | Shanghai Emergency Management Bureau | Includes information from 169 emergency warehouses |
| --- | --- | --- |
| Emergency Shelter | Shanghai Emergency Management Bureau | Includes 117 emergency shelters divided into three classes. |
| Warehouse Club | AutoNavi Open Platform | Includes 27 warehouse club locations |

Future flood inundation scenarios in Shanghai are derived from Yin et al (2020). In their previous work, coastal flood inundation caused by overtopping and dike breaching was simulated via a 2-D flood inundation model (FloodMap-Inertial) with a fine-resolution DEM for three representative return periods (10, 100, and 1000 years) under current and future climate scenarios (RCP 8.5). The study considered climatically driven absolute SLR by using probabilistic, localized SLR projections at the Lvsi gauge station located in the Yangtze River Delta, provided by Kopp et al. (2014). This projection accounts for climatic factors such as ice sheet melting, glacier and ice cap melting, and ocean thermal expansion. In addition, FloodMap-Inertial, developed from FloodMap (Yu and Lane, 2006), has been thoroughly tested and applied in Shanghai (Yin et al., 2016a, 2019), showing reliable performance in flood prediction. This model uses a computationally efficient inertial algorithm to solve the 2-D shallow water equations (Bates et al., 2010), using the Forward Courant-Friedrichs-Lewy (CFL) condition for the calculation of time steps. A complete description of the model structure and parameterization can be found in Yu and Lane (Yu and Lane, 2006). In this study, we focused on the 2030 scenarios with 100- and 1000-year return periods under the RCP 8.5 scenario. The RCP 8.5 scenario represents high radiative forcing and worst-case climate impacts. Thus, these two future scenarios represent extreme flood inundation. The 2030 projections are the closest to the present, making them relevant for near-term planning.

The road network data are from the 2013 Shanghai Traffic Navigation GIS dataset. Data are included on approximately 243,000 road sections with attributes such as road name, type, function, direction, and length. In accordance with the "Technical Standards for Highway Engineering of the People's Republic of China (JTG B01-2003)", the roads are divided into five grades: superhighways, highways, main roads, secondary roads and branch roads.

Shanghai's demographic data were collected during the Sixth National Population Census of China, which utilized a household-level survey. The dataset provides the most detailed information on demographics at the basic administrative level—community or village. According to these data, we calculated the elderly population (65 years old or older) in each community.

The emergency warehouse data for this study are from the Shanghai Emergency Management Bureau. We obtained location information for 169 emergency warehouses. After filtering by facility name, 25 warehouses were identified as potential Emergency Reserve Warehouses (ERWs) that could provide daily living supplies for 281K affected people. Based on surveys and the Standard for the Construction of Relief Goods Reserve Warehouses (Ministry of Civil Affairs of the People's Republic of China, 2009), it was assumed that city-level (Level 1) warehouses can meet the basic needs of 200K affected people, district-level (Level 2) warehouses can meet the needs of 5K people, and township-level (Level 3) warehouses can meet the needs of 3K people.

The data for the emergency shelters used in this paper are from the statistics of the Shanghai Emergency Management Bureau[1]. Data on 117 emergency shelters are included, with attributes such as name, capacity, and level. The shelters are divided into three classes: class 1, class 2, and class 3. Generally, class 1 and class 2 are fixed emergency shelters, class 1 facilities have the capacity to hold more than 5K people, and class 2 facilities can accommodate 1K to 5K people. Considering the nature of coastal flooding, 74 shelters with indoor venues, such as schools, could be Emergency Flood Shelters (EFSs). These EFSs can accommodate approximately 330K people.

The warehouse club data used in this study were obtained from the Points of Interest (POI) dataset provided by the AutoNavi Open Platform. Warehouse clubs are large-scale retail facilities that combine inventory storage and retail operations within a shared space, typically exceeding 10,000 m$^2$ in floor area. Currently, five major operators manage a total of 27 warehouse clubs across Shanghai. For modeling purposes, it was assumed that each warehouse club can supply basic emergency needs for 5K individuals.

### 3.3. Results

### 3.3.1. GIS-based results

Figure 2 illustrates the spatial distributions of ERWs, EFSs and activated EFSs across Shanghai, as well as the affected areas in 100- and 1000-year coastal flood scenarios. Table 2 provides statistical information regarding emergency facilities and disaster situations in two coastal flood scenarios, including the number of available facilities, number of activated EFSs, number of affected communities and population of affected elderly, etc.

Spatially, EFSs are predominantly concentrated in central urban districts, whereas ERWs exhibit a more dispersed pattern. During the 100-year flood scenario, all 25 ERWs and 71 EFSs (96% of the total) remain available, with relatively low spatial exposure risk. The available ERWs have the capacity to provide daily supplies for 281K individuals, and the available EFSs can accommodate up to 313,299 elderly people, accounting for 95% of the total capacity of EFSs. The impact of flooding is most pronounced in areas such as Chongming Island and Baoshan, Huangpu, and Xuhui districts. In this scenario, there are 562 affected communities with approximately 145K exposed elderly individuals. Thus, only 26 EFSs need to be activated to accommodate the impacted elderly population. The activated EFSs, primarily located in the central urban and northern areas of Shanghai, have a total capacity of accommodating approximately 146k individuals, representing 47% of the overall available shelter capacity.

As expected, the 1000-year flood scenario involves more extensive and severe flood inundation in the city. A total of 1,820 communities are exposed to coastal flooding, with approximately 534K elderly individuals in need of relocation to EFSs. In terms of critical facilities, only 21 ERWs and 61 EFSs (82% of the total) are available, indicating a higher spatial exposure risk. The available supplies can meet the needs of 265K people (50% of the total demand), and the available EFSs have the capacity to accommodate 280,919 individuals. Thus, all 61

---

[1] https://gfdy.sh.gov.cn/yjbncs/

available EFSs need to be activated to accommodate the affected elderly population. Only 53% of the total affected elderly individuals can be accommodated at these EFSs.

**Table 2.** Statistics for emergency facilities and disaster situations in two coastal flood scenarios.

| Flood scenarios | Available ERWs | | Available EFSs | | Activated EFSs | | Affected communities | Affected elderly |
|---|---|---|---|---|---|---|---|---|
| | Number | Stock | Number | Capacity | Number | Capacity | Number | Number |
| 100-year | 25 | 281,000 | 71 | 313,299 | 26 | 145,981 | 562 | 145,197 |
| 1000-year | 21 | 265,000 | 61 | 280,919 | 61 | 280,919 | 1,820 | 534,079 |

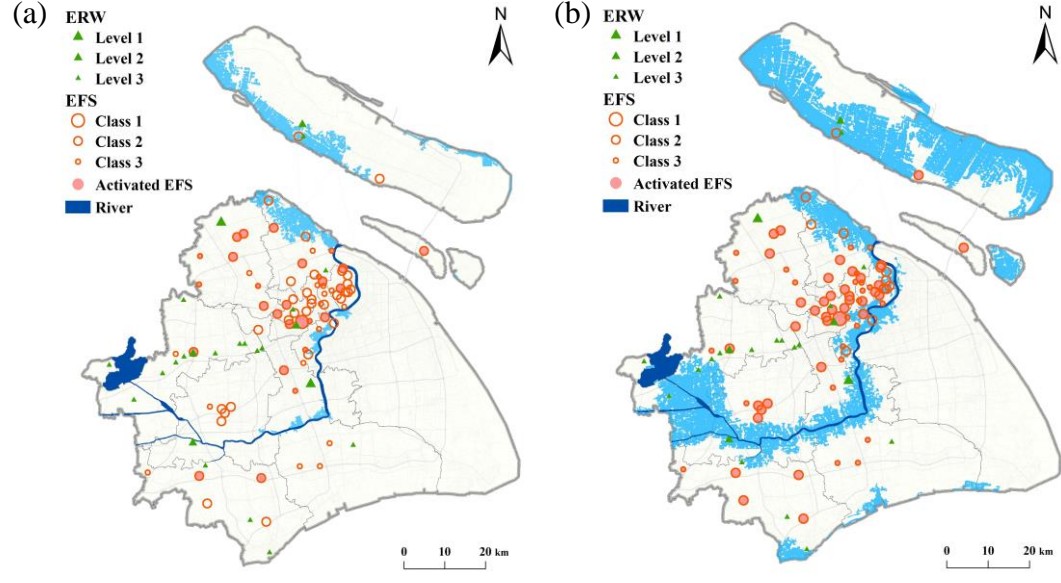

**Figure 2.** Spatial distributions of the emergency reserve warehouses (ERWs) and emergency flood shelters (EFSs) in 100-year (a) and 1000-year (b) flood scenarios in 2030. (The blue-shaded areas represent the flood inundation zone)

### 3.3.2. Resource allocation in two coastal flood scenarios

For the 100-year flood scenario, the total amount of available supplies stockpiled in Shanghai can cover the needs of all affected elderly individuals (Table 2). Therefore, a single-objective model is employed to determine the resource allocation scheme in this scenario. The resource allocation network comprises 25 available ERWs and 26 activated EFSs in total. In terms of the main model parameter, the unit transportation cost ($c_{ij}$) is assumed to be 1 RMB to prevent data unavailability issues. The results indicate that activating 17 ERWs could meet the needs of all elderly people at 26 EFSs. The relative transportation cost is 22,764 RMB. Figure 3 and Table 3 show the resource allocation scheme and the service capacity, respectively, of the activated ERWs in the 100- and 1000-year flood scenarios in 2030.

As shown in Fig. 3a, the demand for activated EFSs can almost entirely be provided by the nearest ERW. The city-level ERW in Jiading District (No. 24) and the branch warehouse in Minhang District (No. 25) play prominent roles in emergency resource provision. In contrast, the ERWs located in the western region of Shanghai are mostly inactive. For example, the city-level ERW and branch warehouse provide emergency resources for 8 and 11 EFSs in Shanghai, accounting for 28.6% and 39.6% of the total demand, respectively (Table 3). Furthermore, Figure 3a indicates that EFSs in the central areas of Shanghai demonstrate significantly higher demand than those in the suburbs; thus, the branch warehouse (No. 25), which is near downtown, provides most of the supplies. The results also show a cross-river supply route from the branch warehouse to an EFS (No. 26) located on a nearshore island; this route is the longest route identified in the scenario.

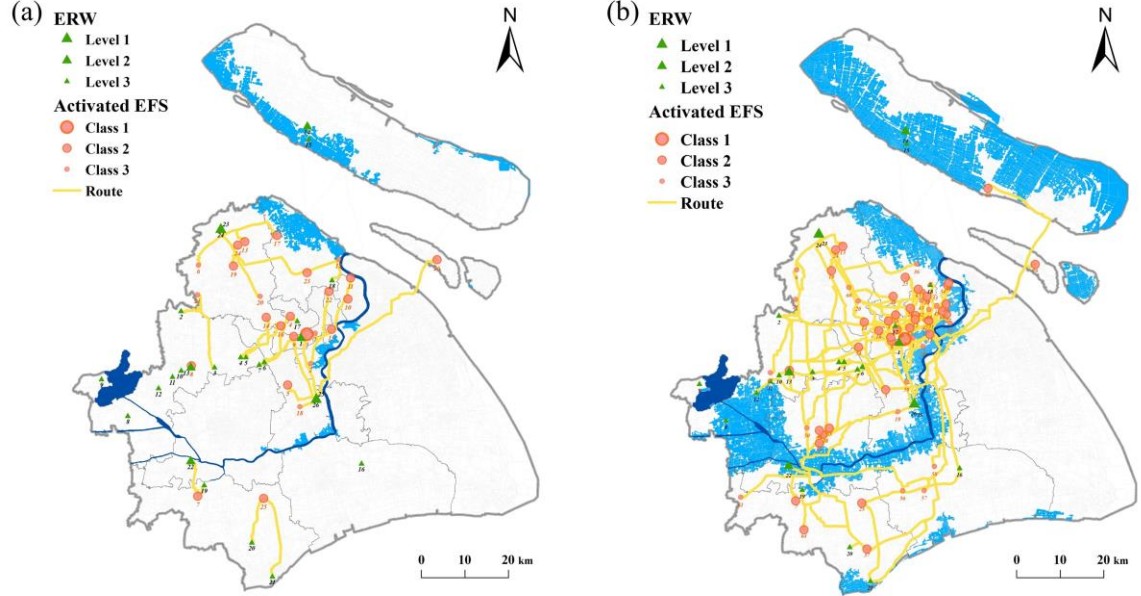

**Figure 3.** Resource allocation scheme for the 100-year (a) and 1000-year (b) flood scenarios in 2030.

**Table 3.** Service capacity of the activated ERWs in two coastal flood scenarios.

| Level | ID | 100-year | | 1000-year | |
|---|---|---|---|---|---|
| | | EFSs served | Supplies ($\times 10^2$) | EFSs served | Supplies ($\times 10^2$) |
| Level 1 | 24 | 8 (30.8%) | 416 (28.6%) | 20 (32.8%) | 1000 (35.6%) |
| | 25 | 11 (42.3%) | 574 (39.5%) | 34 (55.7%) | 1000 (35.6%) |
| Level 2 | 1 | 1 (3.8%) | 50 (3.4%) | 1 (1.6%) | 50 (1.8%) |
| | 13 | 1 (3.8%) | 50 (3.4%) | 5 (8.2%) | 50 (1.8%) |
| | 22 | 1 (3.8%) | 14 (1.0%) | 1 (1.6%) | 50 (1.8%) |
| | 23 | 1 (3.8%) | 50 (3.4%) | 2 (3.3%) | 50 (1.8%) |
| Level 3 | 2 | 1 (3.8%) | 30 (2.1%) | 4 (6.6%) | 30 (1.1%) |
| | 3 | 1 (3.8%) | 10 (0.7%) | 1 (1.6%) | 30 (1.1%) |
| | 4 | 1 (3.8%) | 30 (2.1%) | 5 (8.2%) | 30 (1.1%) |
| | 5 | 2 (7.7%) | 30 (2.1%) | 1 (1.6%) | 30 (1.1%) |
| | 6 | 1 (3.8%) | 30 (2.1%) | 2 (3.3%) | 30 (1.1%) |
| | 7 | 3 (11.5%) | 30 (2.1%) | 4 (6.6%) | 30 (1.1%) |
| | 10 | - | - | 1 (1.6%) | 30 (1.1%) |

| | | | | |
|---|---|---|---|---|
| 11 | - | - | 1 (1.6%) | 30 (1.1%) |
| 12 | - | - | 1 (1.6%) | 30 (1.1%) |
| 16 | - | - | 2 (3.3%) | 30 (1.1%) |
| 17 | 1 (3.8%) | 30 (2.1%) | 5 (8.2%) | 30 (1.1%) |
| 18 | 1 (3.8%) | 30 (2.1%) | 1 (1.6%) | 30 (1.1%) |
| 19 | 1 (3.8%) | 30 (2.1%) | 1 (1.6%) | 30 (1.1%) |
| 20 | 1 (3.8%) | 30 (2.1%) | 1 (1.6%) | 30 (1.1%) |
| 21 | 1 (3.8%) | 20 (1.4%) | 2 (3.3%) | 30 (1.1%) |

Note: The proportion of EFSs served to the total number of activated EFSs and the proportion of supplies provided to the total demand are given in parentheses.

In terms of the 1000-year flood scenario, the total available supplies in Shanghai are insufficient to meet the demand of affected elderly people. To ensure equity in the allocation of limited supplies, a bi-objective model is employed to determine the resource allocation scheme. This scenario involves the allocation of supplies from 21 available ERWs to meet the basic needs of 265K affected people at 61 activated EFSs; these EFSs can accommodate approximately 281K of the affected elderly individuals. The minimum rate of satisfaction $\omega$ for each EFS is set to 1%. This bi-objective mathematical model is solved via the nondominated sorting genetic algorithm II (NSGA-II), which is used to obtain a Pareto optimal solution in multiobjective optimization problems (Deb et al., 2002). NSGA-II is an advanced multiobjective evolutionary algorithm that maintains population diversity across generations through nondominated sorting and promotes a uniform distribution of solutions along the Pareto front using a crowding distance measure. The NSGA-II algorithm is widely used in selected combinatorial optimization problems and has the advantages of fast convergence speed, low computational complexity, and high robustness (Ma et al., 2023; Verma et al., 2021). The corresponding algorithm settings for the solution in this study are shown in Table 4.

**Table 4**. Optimization parameter settings for NSGA-II

| Parameter | Population size | Maximum number of iterations | Pareto fraction | Crossover probability |
|---|---|---|---|---|
| Value | 500 | 3,000 | 0.4 | 0.8 |

Figure 4 presents the Pareto front for the resource allocation scheme obtained with the proposed bi-objective framework. This Pareto front depicts the relationship between the relative transportation cost and the maximum rate of unsatisfied demand, with a total of 200 nondominated solutions. The resource allocation schemes that prioritize equity, resulting in low maximum degrees of unsatisfied demand, correspond to solutions that are plotted close to the x-axis. Conversely, the resource allocation schemes that prioritize efficiency, resulting in lower relative transportation costs, correspond to solutions that are plotted close to the y-axis. The results indicate a variety of strategies for decision-makers; the choice of solution depends on the decision-maker's preference between the relative transportation cost and the degree of demand satisfaction.

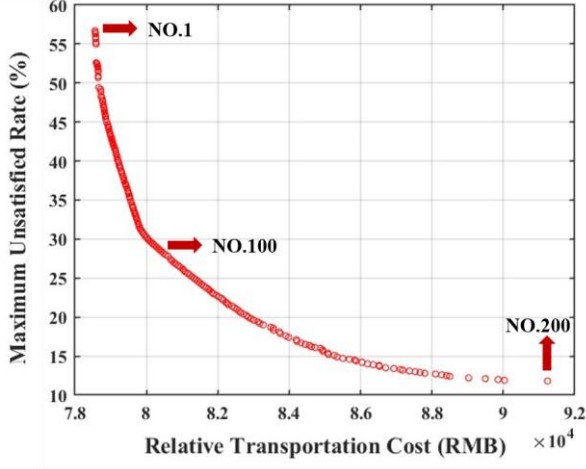

**Figure 4.** Pareto front for relative transportation cost and maximum rate of unsatisfied demand in the 1000-year
flood scenario.

Figure 4 also shows the objective values for three representative solutions (i.e., No. 1, No. 100, and No. 200).
These solutions demonstrate different trade-offs between the relative transportation cost and the maximum rate of
unsatisfied demand. Solution No. 1 yields the lowest relative transportation cost and the highest maximum rate of
unsatisfied demand, totalling 56.7%. Solution No. 200 achieves the lowest maximum rate of unsatisfied demand
(11.8%), and the relative transportation cost is the highest of all values in the solution set. However, solution No.
200 increases the relative transportation cost by 16% compared with that for solution No. 1 but achieves a 44.9%
reduction in the maximum rate of unsatisfied demand. The objective value of solution No. 100 is between the
values of the other two solutions, with a moderate transportation cost and a maximum rate of unsatisfied demand
of 29.8%.

As mentioned above, the choice of resource allocation scheme depends on the decision-maker's priorities. If the
decision maker emphasizes efficiency, solution No. 1 may be adopted, with its low transportation cost. Conversely,
a decision-maker that prioritizes equity can employ, solution No. 200, with a low maximum rate of unsatisfied
demand. Moreover, for a balanced consideration of both efficiency and equity, a middle-ground solution between
No. 1 and No. 200 can be selected, such as efficient solution No. 100.

Figure 3b and Table 3 present the spatial patterns of the resource allocation scheme and the service capacity of
the activated ERWs for solution No. 100 in the 1000-year flood scenario. All available supplies are distributed
(Table 3). The branch ERW (No. 25) provides services to a large number (34) of EFSs, serving 56% of all activated
EFSs. Moreover, Figure 3b reveals that there are relatively few ERWs and that they cannot meet the large demand
for EFSs in the central areas of Shanghai. Furthermore, the results show three cross-river supply routes from the
city-level ERW (No. 24) to EFS No. 26 and from the branch ERW to EFSs No. 26 and No. 39. The former is also
the longest route in this scheme. In addition, the results suggest that the resource demands of 79% of EFSs are
met in solution No. 100. Among the remaining EFSs with unmet demand, EFS No. 36, located in the urban centre,
shows the maximum rate of unsatisfied demand, totalling 29.8%.

### 3.3.3. Analysis of demand variability in the 1000-year flood scenarios

Given the capacity limitations of each EFS, it is impossible to accommodate all affected elderly people in the 1000-year flood scenario. As a result, the activated EFS demand ($Q_j$) in the optimization model includes only the number of elderly people assigned to EFSs in Section 3.3.2. To investigate the results of demand variability, it is assumed that the capacity of each EFS can be moderately adjusted to accommodate more elderly people and that each EFS's demand is increased by 10%, 50%, and 100% (scenario 1, scenario 2, and scenario 3, respectively). The other parameters in the bi-objective optimization model and in the NSGA-II algorithm are consistent with those in Section 3.3.2 (baseline scenario). The Pareto front for the four scenarios is obtained, and the results are presented in Fig. 5. Table 5 lists the objective values of the endpoint solutions of the Pareto frontiers.

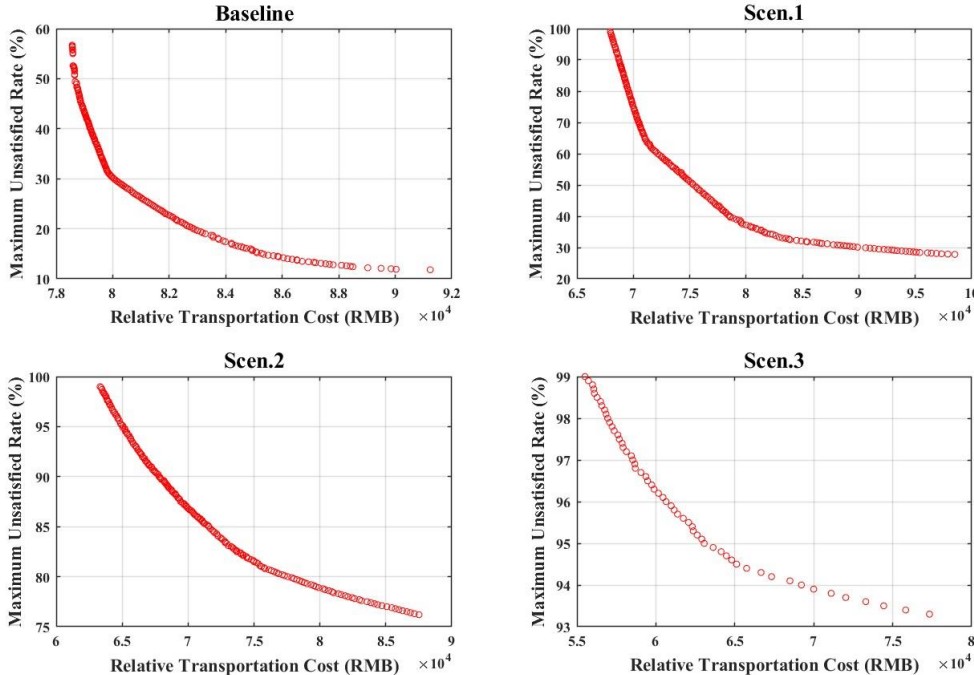

**Figure 5.** Pareto fronts for the relative transportation cost and maximum rate of unsatisfied demand in the four scenarios.

**Table 5.** Objective values of the endpoint solutions of the Pareto fronts in the four scenarios.

| The efficient solution | | Relative transportation cost (RMB) | Maximum unsatisfied rate (%) |
|---|---|---|---|
| Top point | Baseline | 78563.8 | 56.7 |
| | Scen.1 | 67961.0 | 99 |
| | Scen.2 | 63324.5 | 99 |
| | Scen.3 | 55510.5 | 99 |
| Bottom point | Baseline | 91237.4 | 11.8 |
| | Scen.1 | 98501.0 | 27.8 |
| | Scen.2 | 87516.5 | 76.2 |
| | Scen.3 | 77328.8 | 93.3 |

439

In a comparison of the results for the four scenarios, Figure 5 shows that the baseline scenario yields the highest satisfaction rate, with maximum rates of unsatisfied demand ranging from 11.8% to 56.7% (Table 5). Scenario 3 results in a significantly higher rate of unsatisfied demand, falling between 93.3% and 99%. This indicates that a fair allocation scheme can be achieved by reducing the gap between supply and demand. Furthermore, the results demonstrate that Scenario 3 yields a better solution than the other scenarios do in terms of relative transportation costs, i.e., a lower value. Moreover, the relative transportation cost declines gradually from the baseline scenario to Scenario 3, potentially due to the expanding gap between supply and demand, which presents challenges for optimizing the maximum rate of unsatisfied demand. Therefore, the relative transportation cost can be further optimized.

Overall, the results show that significant gaps between supply and demand indicate resource shortages, resulting in a high degree of inequity in the allocation plan. The implication is that solely relying on optimal allocation strategies is inadequate to address the inequity issue. Increasing resource inputs is a crucial fundamental approach to alleviate the unfairness related to resource allocation. These findings aid in understanding the level of equity involved in resource allocation decisions.

### 3.3.4. Analysis of Public-Private Collaboration

As previously indicated, government-held emergency supplies are projected to be insufficient to meet the resource demands of all activated shelters under a 1000-year flood scenario in the 2030s. To address this challenge, an exploratory investigation was conducted into the potential of integrating warehouse clubs in Shanghai as supplementary sources of emergency supplies, with the aim of enhancing the responsiveness and resilience of the emergency supply system. In this context, the proposed resource allocation network comprises a total of 21 available ERWs, 27 available WHCs, and 61 activated EFSs. The aggregated supply capacity across these facilities is sufficient to meet the basic resource requirements of all activated shelters.

A single-objective optimization model was employed to determine the optimal distribution strategy for emergency supplies. The results indicate that, despite the occurrence of road disruptions following the flood event, transportation constraints did not significantly impede access to essential supplies for the activated EFSs. The activation of all 21 ERWs and 26 WHCs was identified as an effective strategy to ensure the adequate provisioning of resources to all shelters. Figure 6 presents the spatial resource allocation scheme after incorporating WHCs under the 1000-year flood scenario in 2030, while table 6 outlines the service capacities of the activated ERWs and WHCs within this context.

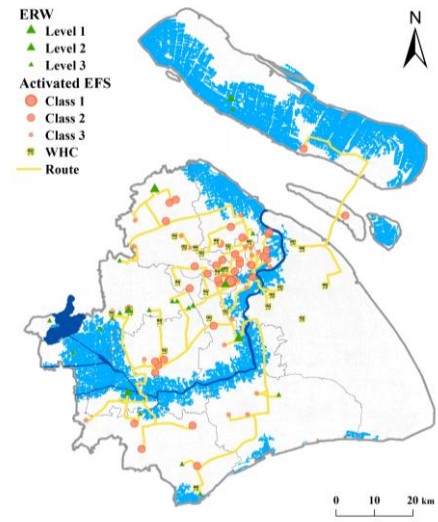


**Figure 6.** Resource allocation scheme for a 1000-year flood scenario in 2030 under public-private collaboration.
**Table 6.** Service capacity of the activated ERWs and WHCs under public-private collaboration

| Facility type | | EFSs served | Supplies ($^2$) |
|---|---|---|---|
| ERWs | Level1 | 21 (34.4%) | 869.19 (30.9%) |
| | Level2 | 9 (14.8%) | 250 (8.9%) |
| | Level3 | 20 (32.8%) | 390 (13.9%) |
| WHCs | | 52 (85.2%) | 1300 (46.3%) |

As illustrated in Fig. 6, warehouse clubs—primarily located on the periphery of the central city—serve as effective
supplementary sources of emergency supplies for shelters situated within the central urban area. This spatial
configuration reduces the transportation burden on government-operated ERWs. Specifically, WHCs supplied a
total of 130k units of emergency resources, accounting for 46.3% of the total supply, to 52 shelters. This indicates
that nearly half of the demand for emergency resources can be met through these facilities. Meanwhile,
government-operated ERWs provided the remaining 54% (Table 6).
The integration of public and private supply chains into a collaborative distribution model not only alleviates
pressure on government-held emergency resources but also enhances the flexibility and responsiveness of the
overall logistics system during disaster response. However, the risk associated with long-distance transportation
remains significant in areas with limited supply infrastructure, such as the southern districts of Shanghai and
Chongming Island. Therefore, future contingency planning should place particular emphasis on pre-positioning
emergency supplies in these vulnerable regions.
**4. Conclusions**
Flood relief logistics planning is a critical component of flood management that directly impacts the livelihood
and security of affected populations. In this study, we presented a comprehensive framework for flood relief
logistics planning using a combination of GIS network analysis and resource allocation optimization models. By
integrating these methodologies, we achieved a synergistic outcome that leverages the strengths of both
approaches. The framework was implemented in Shanghai, China, to explore the availability of ERWs and EFSs
as well as the flood relief logistics plans in future coastal flood scenarios. A number of conclusions can be drawn
from the results. First, the current spatial distribution of ERWs and EFSs in Shanghai shows exposure risks under
extreme coastal floods. Second, while existing facilities can meet elderly needs during a 100-year flood, they
would serve only about half the elderly population in a 1000-year event. Furthermore, although an equity-based
model reduces humanitarian risks under shortages, a supply gap of approximately 6% remains in EFSs. Integrating
private warehouse clubs via public-private partnerships can enhance emergency supply assurance and distribution
efficiency.

Our work can assist emergency managers in better understanding the inadequacies of existing emergency facilities
and highlights the importance of incorporating climate risk information into exhaustive government flood relief
logistics plans. The framework in this study can also be adopted for applications in other coastal cities worldwide.
However, to arrive at more robust conclusions, future studies could focus on the following aspects: 1) Demand
estimation: Given issue of ageing in Shanghai, the elderly population is likely to increase significantly by 2030,
which is highly likely to lead to greater scarcity of shelter resources and supplies. Therefore, future research should
focus on obtaining more detailed data on the elderly population to better understand the spatial distribution and
temporal changes in the affected elderly population. 2) Traffic scenarios: Currently, the model assumes that roads
are closed when the water level reaches 30 cm. Future work should incorporate more complex traffic scenarios,
such as variable speeds at which vehicles can safely navigate flooded areas, to better simulate real-world
conditions. 3) Model validation: This study has not yet incorporated a formal validation of the proposed models.
Comparing model outputs with historical flood event data or established decision models would provide a more
comprehensive validation and enhance the robustness of the framework. Future work should prioritize these
comparisons to improve the model.

Furthermore, in this study, the disaster situation was explored in ArcGIS, and the resource allocation models was
developed in MATLAB. Therefore, future efforts could focus on developing comprehensive decision-support
systems and large models that integrate disaster assessment with relief resource allocation models. Such systems
can offer predictive analytics and scenario-based simulations, enabling proactive decision-making. By filling these
research gaps, researchers can contribute to optimizing effective flood relief logistics planning in the future,
providing more resilient and adaptive emergency responses in coastal cities worldwide.

Code and Data Availability
The optimization models for resource allocation in this study were developed utilizing MATLAB's Optimization
Toolbox with MatlabR2022a. The disaster scenarios were visualized and analyzed within ArcGIS. The source
code for our models is available upon request from the corresponding author. The emergency shelter data for
Shanghai can be publicly accessed (https://gfdy.sh.gov.cn/yjbncs/). Other datasets are not open-source. We have
obtained all necessary permissions for the use of these datasets in our research.

Author Contributions
Jie Yin and Pujun Liang designed the study. Jie Yin, Dandan Wang, and Yuhan Yang collected and provided the
dataset. Pujun Liang conducted the data pre-processing, designed the model and wrote the manuscript. Yi Lu, Dan
Gao and Jianfeng Mai contributed to manuscript revision.

Competing Interests
The contact author has declared that none of the authors has any competing interests.

Disclaimer
Publisher's note: Copernicus Publications remains neutral with regard to jurisdictional claims made in the text,
published maps, institutional affiliations, or any other geographical representation in this paper. While Copernicus
Publications makes every effort to include appropriate place names, the final responsibility lies with the authors.

Financial Support
This paper was supported by the National Natural Science Foundation of China (Grant No. 42371076) and
Shanghai Pilot Program for Basic Research (Grant No: TQ20240209).

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
