# Peer review of "Flood relief logistics planning for coastal cities: a case study in"

_Natural Hazards and Earth System Sciences, 2024_

## Author Comment (AC1)

**Responses to Reviewers Reviewer #1:**

The authors presented a study on flood relief logistics planning based on Geographic Information System (GIS) analysis and resource allocation optimization models in the Shanghai area. They explored the effectiveness and fairness of resource distribution in managing flood crises under 100-year and 1000-year flood scenarios. They found that the current capacities of emergency flood shelters (EFSs) and emergency reserve warehouses (ERWs) are adequate for a 100-year flood but insufficient for a 1000-year flood scenario, and highlighted the need for greater resource investments to address potential shortages. In general, this study is interesting and has practical significance. Most parts of the manuscript are well structured and expressed. This study would be helpful for the community of disaster management and urban planning. However, the current manuscript needs a major revision before it is published in this journal.

We greatly appreciate the invaluable and constructive feedback provided by Reviewer #1. Our responses are highlighted in blue italic. We have acted upon all the points raised. The comments were very useful in improving the overall quality and readability of the manuscript.

**Comment 1:** The paper presents a well-integrated framework for flood relief logistics that combines Geographic Information Systems (GIS) and optimization models. However, the validation of these models is primarily limited to a case study without comparisons to actual event data or established models. Comparing the proposed model outputs with historical flood events or the results from established models would significantly enhance the manuscript's robustness. I suggest the authors to add a discussion in the last part.

Thank you for your valuable feedback on our paper. The issue of model validation you pointed out is indeed important. We have added a discussion in the final section highlighting the need for such validation and indicating the direction for future work. Specifically, we have added the sentence as follows:

line 445-447: 'Third, this study has not yet included formal validation of the proposed models. Future work should prioritize comparing model outputs with historical flood event data or other models to enhance the robustness.'

**Comment 2:** The manuscript briefly mentions specific details about the optimization methods used, such as the NSGA-II algorithm and parameter setting withoutin-depth explanations. Providing detailed descriptions of these methods would enhance the reproducibility of the paper and offer a clearer understanding for readers with specialized knowledge.

Thank you very much for the suggestion. We have added the sentence as follows:

Line 347-353: 'NSGA-II (Non-dominated Sorting Genetic Algorithm II) is an advanced multiobjective evolutionary algorithm that maintains population diversity across generations through non-dominated sorting and promotes uniform distribution of solutions along the Pareto front using a crowding distance measure. As a mature and widely applied method for solving complex multiobjective problems, NSGA-II is included in the MATLAB Optimization Toolbox. The parameters used in this study are summarized in the following table:'

 Table 4. NSGA-II Parameters Table

| Parameter | Population
Size | Maximum
Number of
Iterations | Pareto
Fraction | Crossover
Probability |
|-----------|--------------------|------------------------------------|--------------------|--------------------------|
| Value     | 500                | 3,000                              | 0.4                | 0.8                      |

**Comment 3:** More comprehensive details regarding the data sources used in this study would be beneficial. Clarifying the availability and accessibility of these data for other researchers or planners, as well as disclosing any proprietary or restricted data, would enhance the transparency and applicability of the research.

Thank you very much for your comment. Regarding the data sources used in this study, we have provided more comprehensive details. The emergency shelter data for Shanghai, which can be publicly accessed, was provided by the Shanghai Emergency Management Bureau and is available at https://gfdy.sh.gov.cn/yjbncs/. This information has been noted in line 280 of the revised manuscript.

Specifically, the remaining datasets include:

- 1) The future flood inundation scenarios in Shanghai under the climate scenarios used in this study were previously established by the authors (Yin et al., 2020).
- 2) The community census data and road network data for Shanghai were provided by Shanghai Municipal Bureau of Statisticsa and Key Laboratory of the Ministry of Education at East China Normal University.
- 3) The data for the emergency warehouse locations were supplied by our collaborating institution, the Shanghai Emergency Management Bureau.

These datasets are not open-source. We have ensured that the necessary permissions have been obtained for their use in this study.

**Comment 4:** The manuscript mostly cited is relatively old. It is recommended to add more recent researches that would update and enhance its relevance to current disaster management and urban planning challenges.

Thank you very much for your comment. We added the references to include recent studies in the fields of disaster management and urban planning as follows:

Line 55-60: 'Moreover, The New Urban Agenda outlines actions to strengthen cities' capacities to reduce disaster risks and mitigate their impacts (Habitat III, 2017). The Making Cities Resilient 2030 (MCR2030) initiative advocates for incorporating climate risk projections into disaster risk reduction and resilience strategies (UNDRR, 2022). Yin et al. (2024) demonstrate the improved performance of risk-informed, strategic evacuation planning in advance of coastal flooding. Additionally, the IPCC highlights the critical role of humanitarian responses and local disaster management in disaster risk reduction (IPCC, 2023). '

We also added some references in revised manuscript as follows:

*Lines 33-34: 'Over the past two decades, the number of major flood events has more than double, claiming approximately 1.2 million lives and impacting over 4.03 billion people. (Mizutori and Guha-Sapir, 2020).'*

Lines 45-46: 'Disaster risk management systems face increasing challenges in adapting to evolving risk profiles (IPCC, 2023).'

**Comment 5:** The language of this paper needs to be further refined since some language expressions are not accurate, and the expression in some places is too redundant.

Thank you very much for your suggestion. In the updated version, we refined the manuscript to correct redundant expressions and inaccuracies throughout the text.

**Comment 6:** The captions of figures and table can remove "the". Data sources and model parameter variables are best represented by tables.

Thank you very much for your comment. We appreciate your valuable suggestion. In the revised manuscript, we have removed "the" from all figure and table captions and added additional table as follows.

| Data Type                   | Source                                                                             | Description                                                                                                                |  |
|-----------------------------|------------------------------------------------------------------------------------|----------------------------------------------------------------------------------------------------------------------------|--|
| Flood
Inundation
Maps | Yin et al. (2020)                                                                  | Simulated coastal flood inundation
scenarios for 100-year and 1000-year
return periods under RCP 8.5 scenario.       |  |
| Road Network                | Key Laboratory of the
Ministry of Education at East
China Normal University. | Comprises approximately 243,000 road
sections with attributes including name,
type, function, direction, and length. |  |
| Demographic                 | Shanghai Municipal Bureau
of Statistics                                         | Detailed demographic information at the community level.                                                                   |  |
| Emergency
Warehouse      | Shanghai Emergency
Management Bureau                                            | Includes 169 emergency warehouses.                                                                                         |  |
| Emergency
Shelter        | Shanghai Emergency
Management Bureau                                            | Includes 117 emergency shelters divided into three classes.                                                                |  |

Table 1. Data Sources Information.

**Table 4. NSGA-II Parameters Table**

| Parameter | Population
Size | Maximum
Number of
Iterations | Pareto
Fraction | Crossover
Probability |
|-----------|--------------------|------------------------------------|--------------------|--------------------------|
| Value     | 500                | 3,000                              | 0.4                | 0.8                      |

**Comment 7:** The authors selected two scenarios of 100-year and 1000-year for comparison. Does it fully consider the differences in other scenarios ? For example, 500-year, will it affect the results ? It is suggested to add some discussion.

Thank you very much for your comment. We chose these two scenarios primarily because they represent situations where supply either exceeds or falls short of demand. In the case of a 1000-year flood scenario, where supply is insufficient, decision-makers may need to consider the fairness of resource allocation. Therefore, we proposed a bi-objective allocation model to address this issue and provide guidance for decision-makers.

A 500-year flood event would involve different population needs, leading to different allocation outcomes. However, given that the 100-year and 1000-year scenarios adequately represent the supply-demand dynamics that decision-makers encounter in resource allocation, we focused on these two scenarios to demonstrate the application of our allocation model.

While we did not explore the results for other scenarios in this paper, future research involving the optimization of emergency resource locations for multiple recurrence periods will consider more scenarios.

---

## Author Comment (AC2)

**Responses to Reviewers**

**Reviewer #2:**

In the context of climate change, more and more extreme events are occurring in coastal cities, increasing disaster risk. Disaster emergency rescue needs are greater, and how to allocate the available rescue resources is an issue worthy of further study. The author combines resource status with allocation management, considering the efficiency of resource allocation and the equity between regions. The flood relief logistics planning framework can be used to guide the allocation of emergency relief materials. Shanghai is a high-risk area for flooding and needs emergency rescue. This paper presents a comprehensive framework for flood relief logistics planning using a combination of GIS network analysis and analysis. The resource allocation optimization model projects the 100-year and 1000-year emergency rescue logistic allocation scenarios in the study area Shanghai, which has important scientific and practical significance for the emergency rescue for Shanghai.

It is suggested accepted with minor revision.

Before the manuscript to be accepted for published, some points should be made clearer.

*We greatly appreciate the invaluable and constructive feedback provided by Reviewer #2. Our responses are highlighted in blue italic. We have acted upon all the points raised. The comments were very useful in improving the overall quality and readability of the manuscript.*

**Comment 1:** Line 80~84: Is the motivation for this study due to the lack of research, or the lack of consideration of future climate change scenarios and supplies shortages?

*Thank you very much for your valuable comment. The motivation for this study stems from three key gaps in the literature. First, while disaster logistics has been widely studied, there is limited research specifically focused on flood scenarios. Second, these existing studies are based on historical flood scenarios, without considering the potential increased risks posed by extreme weather events under future climate scenarios. Last, these existing studies focus on optimizing efficiency in resource distribution, there is relatively little research on ensuring fairness in the allocation process.*

*We have revised some sentences as follows:*

*Line 89-91: 'While considerable research has been performed in the field of disaster relief logistics, less attention has been given to flood relief logistics modeling. Only a few studies consider the impact of floods on resource distribution logistics, particularly the disruption caused by the inundation of emergency facilities and roads.'*

*Line 99-102: 'However, it is noteworthy that these existing studies are primarily based on*

*historical flood scenarios and do not adequately consider the potential increased risks posed by extreme flood events under future climate scenarios. Moreover, a majority of these studies focus on optimizing the efficiency of resource distribution, while there is relatively little research dedicated to ensuring fairness in the allocation process.'*

**Comment 2:** Line 16~18: "Considering the fairness of resource allocation, a biobjective allocation model that minimizes the total transportation cost and maximum unsatisfied rate is developed." Why maximum unsatisfied rate?

*Thank you very much for your important question. In situations where resources are insufficient, some regions may inevitably receive less than the required amount, leading to an unmet demand rate for each area. The maximum unsatisfied rate refers to the highest unmet demand rate among these regions. By minimizing this rate, we aim to reduce disparities in unmet demand across regions, ensuring that no area experiences extreme shortages. This objective promotes fairness by preventing any region from bearing a disproportionately high burden of unmet needs. We have included an explanation of this concept in the revised version as follows:*

*Line 217-219: 'The maximum unsatisfied rate refers to the highest unmet demand rate among these regions. By minimizing this rate, we aim to reduce disparities in unmet demand across regions, ensuring that no area experiences extreme shortages.'*

**Comment 3:** Line 110~113: When supply exceeds demand, emergency managers tend to focus on maximizing efficiency to optimally allocate resources. Lack of supply should be considered more in efficiency. It should be said that when supply is plentiful, considering efficiency alone is enough. When supply is shortage, attention should be paid to both efficiency and equity. But is it a scientific and technical issue or a managing issue?

*We agree with the comment. We have removed the statement in line 110-113 and added the sentence as follows:*

*Line 211-213: 'When supply is insufficient to meet demand, emergency managers should ensure that resources are distributed fairly across regions to prevent humanitarian inequalities caused by unbalanced allocation.'*

*Regarding your question on whether this is a scientific and technical issue or a management issue, we believe that in practice, the decision to balance efficiency and fairness is largely a management decision. Emergency managers must weigh these objectives based on factors such as the severity of the disaster and the availability of resources. However, addressing this balance effectively often requires complex mathematical optimization, which depends on scientific and technical support.*

**Comment 4:** Line 300 and 324: The cyan area in Figures 2 and 3 should be explained (legend).

*Thank you very much for the suggestion. We have added an explanation of the cyan area in the legend in the data source section as follows:*

*line 275-277: 'Based on surveys and the Standard for the Construction of Relief Goods Reserve Warehouses (Ministry of Civil Affairs of the People's Republic of China, 2009), we categorized the warehouses into three levels: city-level (Level 1) warehouses, which can meet the basic needs of 200,000 affected people; district-level (Level 2) warehouses, which can serve 5,000 people; and township-level (Level 3) warehouses, which can support 3,000 people.'*

**Comment 5:** Line 409~439: It is suggested that the conclusion condenses more definite points.

*Thank you for your valuable feedback regarding the conclusion of our manuscript. We have revised the last two paragraphs of the conclusion section as follows:*

*Line 438-454: 'Our work can assist emergency managers in better understanding the inadequacies of existing emergency facilities and highlights the importance of incorporating climate risk informed into exhaustive government flood relief logistics plans. The framework in this study can also be adopted for applications in other coastal cities worldwide. However, to arrive at more robust conclusions, future studies could be directed to the following aspects: First, concerning demand estimation, a more precise methodology should take into consideration the willingness of various individuals affected by flooding to reside in shelters. Second, Future research should incorporate more complex traffic scenarios, such as variable speeds at which vehicles can safely navigate flooded areas, to better simulate real-world conditions. Third, this study has not yet included formal validation of the proposed models. Future work should prioritize comparing model outputs with historical flood event data or other models to enhance the robustness.*

*Furthermore, in this study, the disaster situation is explored in ArcGIS, and the resource allocation models are developed in MATLAB. Therefore, future efforts could focus on developing comprehensive decision-support systems and large models that integrate disaster assessment with relief resource allocation models. Such systems can offer predictive analytics and scenario-based simulations, enabling proactive decision-making. By filling these research gaps, the future of effective flood relief logistics planning can be ensured, providing more resilient and adaptive emergency responses in coastal cities worldwide.'*

---

## Author Comment (AC4)

**Responses to Reviewers**

**Reviewer #2:**

This paper proposes a logistics planning framework for flood relief tailored to coastal cities, with Shanghai serving as a case study. The authors integrate GIS network analysis and resource allocation optimization models to investigate emergency management strategies under different flood scenarios. The framework offers valuable support for decision-making by incorporating geographic and resource allocation data to enhance flood relief efforts. However, the manuscript requires significant revisions. The main concerns are outlined below.

*We greatly appreciate the invaluable and constructive feedback provided by Reviewer #2. Our responses are highlighted in blue italic. We have acted upon all the points raised. The comments were very useful in improving the overall quality and readability of the manuscript.*

**Comment 1:** The paper does not include a description of the flood models used, referencing only Yin et al. (2020). The referenced study covers various flood scenarios across different years. Why does this paper focus solely on the 2030 scenarios with 100- and 1000-year return periods?

*Thank you very much for your valuable comment. We provide a more detailed explanation of the flood models used and justify the selection of these specific scenarios in the revised manuscript. We have added some sentences as follows:*

*Line 265-270: 'In addition, FloodMap-Inertial developed from FloodMap (Yu & Lane, 2006), has been thoroughly tested and applied in in Shanghai (Yin et al., 2015, 2019), showing reliable performance in flood prediction. This model utilizes a computationally efficient inertial algorithm to solve the 2-D shallow water equations (Bates et al., 2010), using the Forward Courant-Friedrichs-Lewy (CFL) Condition for the calculation of time steps. A complete description of the model structure and parameterization can be found in Yu and Lane (2011).'*

*Line 271-274: ' In this study, we focused on the 2030 scenarios with 100- and 1000-year return periods under the RCP 8.5 scenario. The RCP 8.5 scenario represents high radiative forcing and worst-case climate impacts. Thus, these two future scenarios represent extreme flood inundation. The 2030 projections are the closest to the present, making them relevant for near-term planning.'*

*The flood inundation results used in this study are shown in the following figure.*

[Figure]

*Figure. Projected flood inundation under future scenarios (RCP 8.5) in 2030. One hundred‑year flood (a) and One thousand‑year flood (b). (Yin et al. ,2020).*

**Comment 2:** The population data utilized in the analysis is from 2010. Given the aging population trend between 2010 and 2030, how might this demographic shift affect the analysis results? Would it significantly impact the findings?

*Thank you very much for your valuable comment. Considering the aging trend between 2010 and 2030, demographic shifts could affect the analysis results. In fact, we do not have specific projections data of the elderly population at the community level. We only can refer to the overall projections provided by the Shanghai Statistics Bureau. According to their reports, the projected elderly population in Shanghai for 2030 is expected to reach 4.8 million, an increase of approximately 106% compared to 2010.*

*Assuming the elderly population in affected communities grows at the same rate, under a 100-year flood event, the number of affected elderly individuals would be approximately 299,138 and under a 1000-year flood event, it would rise to around 1.1 million—more than double the number affected in 2010. When comparing these figures with the current capacity of emergency flood shelters (EFSs) and the supplies stored in emergency reserve warehouses (ERWs), we find that while the capacity of EFSs is adequate for a 100-year flood event, ERWs would not be sufficient. For a 1000-year flood event, neither EFSs nor ERWs would suffice to meet the needs of the elderly population.*

*While the simple analysis as mentioned suggests that an increase in the elderly population could affect the results, it assumes that the growth rate of the elderly population in affected communities is the same, which may not be the case in reality. Due to the lack of detailed spatiotemporal projections of the elderly population in affected communities, we cannot draw*

*definitive conclusions about the impact. Consequently, the final results can only suggest a high probability that the increase in the elderly population by 2030 will lead to greater shortages in shelter resources and supplies in two costal flood scenarios.*

*As a result, in the revised manuscript, we have added a discussion section to address these limitations. Specifically, we propose that future research should focus on obtaining more detailed spatiotemporal data on the elderly population to better understand the spatial distribution and temporal changes in the affected elderly population. This would be beneficial to refine the social vulnerability component of the risk assessment and further optimize resource allocation strategies.*

*We have revised some sentences as follows:*

*Line 457-461: 'However, to arrive at more robust conclusions, future studies could be directed to the following aspects: 1) Demand Estimation: Given the aging issue in Shanghai, the elderly population is likely to increase significantly by 2030, leading to a high probability of greater scarcity in shelter resources and supplies. Therefore, future research should focus on obtaining more detailed data on the elderly population to better understand the spatial distribution and temporal changes in the affected elderly population.'*

**Comment 3:** In Equation 1, the number of affected individuals is estimated based on the proportion of flooded areas. However, is there a valid linear relationship between the number of people affected and the flooded area? Further justification or discussion of this assumption is needed.

*Thank you very much for your valuable comment. Currently, the available data is at the community level, and we do not have more detailed the elderly population data. Therefore, to estimate the number of affected elderly individuals within each community, this study assumes a uniform spatial distribution of the elderly population. The number of affected elderly individuals is calculated using Equation 1.*

*In the revised manuscript, we have further clarified this assumption as follows:*

*Line 155-158: 'Additionally, due to the lack of more detailed spatial distribution data for the elderly population, this study assumes a uniform distribution within the community. The number of elderly individuals in each affected community can be calculated using Equation 1.'*

**Comment 4:** Line 90 mentions that previous studies did not consider flood scenarios under climate change. Does the 2030 flood scenario used in this study genuinely reflect a climate change scenario, and if so, how?

*We thank the reviewer for pointing out the need for clarity regarding the consideration of*

*climate change in the flood scenarios used in our study. The 2030 flood scenario employed in this study reflects a climate change scenario, as it incorporates projections of sea level rise under the RCP 8.5 scenario. Specifically, the flood inundation scenarios are derived from Yin et al. (2020), which included climatically driven absolute sea level rise projections provided by Kopp et al. (2014). These projections consider factors such as ice sheet melting, glacier and ice cap melting, and ocean thermal expansion.*

*To further clarify this point, we have revised the manuscript to explicitly state that the flood scenarios considered in our study incorporate climate change impacts. We have revised some sentences as follows:*

*Line 257-263: 'Future flood inundation scenarios in Shanghai are derived from Yin et al. (2020). In their previous work, coastal flood inundation caused by overtopping and dike breaching was simulated using a 2-D flood inundation model (FloodMap-Inertial) with a fine-resolution DEM for three representative return periods (10, 100, and 1000 years) under current and future climate scenarios (RCP 8.5). And, the study considered climatically driven absolute sea level rise (SLR) by using the probabilistic, localized SLR projections at the Lvsi gauge station located in the Yangtze River Delta, provided by Kopp et al. (2014). This projection takes into account climatic factors such as ice sheet melting, glacier and ice cap melting, and ocean thermal expansion.'*

**Comment 5:** The application of the bi-objective model in multi-objective optimization is central to this paper. However, the background description of the model is insufficient, particularly regarding the implementation of the NSGA-II algorithm. It is recommended to provide more details on the algorithm's steps and discuss its advantages in practical applications.

*Thank you very much for the suggestion. We agree that a more detailed explanation of the NSGA-II algorithm is necessary. We have added some sentence as follows:*

*Line 363-373: 'This biobjective mathematical model is solved by the NSGA-II (Non-dominated Sorting Genetic Algorithm II) algorithm which is used to obtain Pareto optimal solution in multi-objective optimization problems (Deb et al., 2002). NSGA-II is an advanced multi-objective evolutionary algorithm that maintains population diversity across generations through non-dominated sorting and promotes uniform distribution of solutions along the Pareto front using a crowding distance measure. NSGA-II algorithm is widely used in selected combinatorial optimization problems, with the advantages of fast convergence speed, low computational complexity, and high robustness (Ma et al., 2023; Verma et al., 2021). The corresponding algorithm settings for the solution in this study are shown in Table4.'*

*Table 4. Optimization parameter settings for NSGA-II*

| Parameter | Population Size | Maximum Number of Iterations | Pareto Fraction | Crossover Probability |
|---|---|---|---|---|
| Value | 500 | 3,000 | 0.4 | 0.8 |

**Comment 6:** For extreme flood scenarios, does the model account for time constraints associated with emergency response? How does the model ensure that supplies can be delivered to affected areas in a timely manner?

*Thank you very much for your valuable comment. The model does not explicitly account for time constraint. According to the '14th Five-Year National Comprehensive Disaster Prevention and Mitigation Plan' in China, it is required that basic living needs of affected individuals be met within 12 hours after a disaster occurs. In this study, we primarily consider the city scale, assuming supplies can be delivered within 12 hours. Moreover, in Shanghai, supplies can be distributed within 3 hours. This timeframe is considered acceptable, so no specific time parameters were set.*

*We have also clarified our assumptions in the paper:*

*Line 147-148: 'A8: Any resource allocation within the city can be completed within 12 hours. Notably, the '14th Five-Year National Comprehensive Disaster Prevention and Mitigation Plan' in China stipulates that affected individuals' basic living needs will be met within 12 hours'*

---

## Author Response (AR2)

**Responses to Reviewers**

**Reviewer #2:**

The authors presented a study on future flood relief logistics planning based on GIS analysis and mathematical modelling to develop plans for disaster management. The planning methodology is applied to the case of Shanghai, China, to draw conclusions. They found that the supply levels of EFSs and ERWs vary in different coastal flood scenarios. Based on this, a flood relief logistics planning was developed for different storm surge flood scenarios. In general, this study is interesting and complete, which holds practical significance. Most parts of manuscript are well structured and expressed. This study would be helpful for the natural hazards community. I recommend acceptance with minor revisions, as detailed below:

*We greatly appreciate the invaluable feedback provided by Reviewer #2. Our responses are highlighted in blue italic. We have carefully considered each suggestion and made corresponding revisions in the revised version.*

**Comment 1:** The use of examples in the Introduction is good, but given that the case study is in China, I would expect to see some examples from there as well.

*Thank you for your valuable suggestion. We have incorporated an example from China into the Introduction to better align with the context of our case study. Specifically, we have added the sentence as follows:*

*line 42-45: 'Typhoon Mangkhut, which hit Hong Kong in 2018, generated a record-breaking storm surge that caused widespread damage across the city. At least 458 people were injured, and the direct economic losses amounted to approximately HKD 4.6 billion (Choy and Wu, 2018). '*

**Comment 2:** In Figure 2 and Figure 3, it is necessary to add descriptions of the shaded areas. Clarifying what these shaded zones represent will enhance the readability of the maps.

*Thank you very much for the suggestion. We have added descriptive notes to clarify the shaded areas. Specifically, we have included the following explanatory text at lines 346-347: '(The blue-shaded areas represent the flood inundation zone)'*

**Comment 3:** The analysis based on GIS assessed the effectiveness of EFSs and ERWs under two flood scenarios, but did not specifically address their spatial exposure. A discussion on this would

add depth to the analysis and improve the risk-informed perspective of the study.

*Thank you for this insightful comment. We fully agree that understanding the spatial exposure of emergency facilities is crucial for risk-informed decision-making. In response, we have added a more comprehensive spatial analysis of EFSs and ERWs, highlighting their distribution patterns and exposure levels under different flood scenarios.*

*1)  Line 323-325: 'Spatially, EFSs are predominantly concentrated in central urban districts, whereas ERWs exhibit a more dispersed pattern. During the 100-year flood scenario, all 25 ERWs and 71 EFSs (96% of the total) available, with relatively low spatial exposure risk.'*

*2)  Line 330-332: 'The activated EFSs, primarily located in the central urban and northern areas of Shanghai, have a total capacity of accommodating approximately 146k individuals, representing 47% of the overall available shelter capacity.'*

*3)  Line 336-337: 'In terms of critical facilities, only 21 ERWs and 61 EFSs (82% of the total) are available, indicating a higher spatial exposure risk.'*

**Comment 4:** The article emphasizes flood relief logistics planning in coastal cities to cope with different storm surge flood scenarios in the context of climate change. However, when facing a 1000-year flood scenario, existing government resources are insufficient, whether alternative cooperative schemes such as public-private partnerships can be used to supplement resources could be further discussed.

*We sincerely appreciate this insightful suggestion regarding public-private partnerships in extreme flood scenarios. As duly noted, our original analysis identified critical gaps in government resource capacity during 1000-year flood events. In response, we have added a new subsection titled "Analysis of Public-Private collaboration" (Section 3.3.4)*

*Line 461-494: 'As previously indicated, government-held emergency supplies are projected to be insufficient to meet the resource demands of all activated shelters under a 1000-year flood scenario in the 2030s. To address this challenge, an exploratory investigation was conducted into the potential of integrating warehouse clubs in Shanghai as supplementary sources of emergency supplies, with the aim of enhancing the responsiveness and resilience of the emergency supply system. In this context, the proposed resource allocation network comprises a total of 21 available ERWs, 27 WHCs, and 61 activated EFSs. The aggregated supply capacity across these facilities is sufficient to meet the basic resource requirements of all activated shelters.*

*A single-objective optimization model was employed to determine the optimal distribution strategy for emergency supplies. The results indicate that, despite the occurrence of road disruptions following the flood event, transportation constraints did not significantly impede access to essential supplies for the activated EFSs. The activation of all 21 ERWs and 26 WHCs was identified as an effective strategy to ensure the adequate provisioning of resources to all shelters. Figure 6 presents the spatial resource allocation scheme after incorporating WHCs under the 1000-year flood scenario in 2030, while table 6 outlines the service capacities of the activated ERWs and WHCs within this context.*

[Figure]

*Figure 6. Resource allocation scheme for a 1000-year flood scenario in 2030 under public-private collaboration.*

*Table6. Service capacity of the activated ERWs and WHCs under public-private collaboration*

| | Facility type | EFSs served | Supplies ($\times 10^2$) |
|---|---|---|---|
| | *Level1* | *21 (34.4%)* | *869.19 (30.9%)* |
| *ERWs* | *Level2* | *9 (14.8%)* | *250 (8.9%)* |
| | *Level3* | *20 (32.8%)* | *390 (13.9%)* |
| *WHCs* | | *52 (85.2%)* | *1300 (46.3%)* |

*As illustrated in Figure 6, warehouse clubs—primarily located on the periphery of the central city—serve as effective supplementary sources of emergency supplies for shelters situated within the central urban area. This spatial configuration reduces the transportation burden on*

*government-operated ERWs. Specifically, WHCs supplied a total of 130k units of emergency resources, accounting for 46.3% of the total supply, to 52 shelters. This indicates that nearly half of the demand for emergency resources can be met through these facilities. Meanwhile, government-operated ERWs provided the remaining 54% (see Table 6).*

*The integration of public and private supply chains into a collaborative distribution model not only alleviates pressure on government-held emergency resources but also enhances the flexibility and responsiveness of the overall logistics system during disaster response. However, the risk associated with long-distance transportation remains significant in areas with limited supply infrastructure, such as the southern districts of Shanghai and Chongming Island. Therefore, future contingency planning should place particular emphasis on pre-positioning emergency supplies in these vulnerable regions.'*

**Comment 5:** The conclusion reads more like a discussion. The author emphasizes future research directions. This section needs to be expanded by including research findings.

*Thank you very much for your suggestion. In the updated version, we have added the sentence in Conclusion to better reflect the key findings as follows:*

*Lines 502-508:'A number of conclusions can be drawn from the results. First, the current spatial distribution of ERWs and EFSs in Shanghai shows exposure risks under extreme coastal floods. Second, while existing facilities can meet elderly needs during a 100-year flood, they would serve only about half the elderly population in a 1000-year event. Furthermore, although an equity-based model reduces humanitarian risks under shortages, a supply gap of approximately 6% remains in EFSs. Integrating private warehouse clubs via public-private partnerships can enhance emergency supply assurance and distribution efficiency.'*

*We have also updated the abstract to reflect the main findings of the study:*

*Lines 2-8:'The case study indicates that the current spatial distribution of Emergency Reserve Warehouses (ERWs) and Emergency Flood Shelters (EFSs) in Shanghai may be vulnerable to extreme flood events. Under a 1000-year coastal flood scenario, the existing emergency resources are insufficient to meet the needs of the affected elderly population. In situations of resource scarcity, reducing the maximum unsatisfied rate can help improve the equity of resource allocation.*

*Furthermore, incorporating private warehouse clubs (WHCs) into government emergency logistics through public-private collaboration could reduce governmental burden and improves system efficiency and resilience.'*